# Hibernation induces widespread transcriptional remodeling in metabolic tissues of the grizzly bear

Heiko T. Jansen [1,5], Shawn Trojahn [2,5], Michael W. Saxton [2,5], Corey R. Quackenbush[2], Brandon D. Evans Hutzenbiler[1,3], O. Lynne Nelson[4], Omar E. Cornejo[2], Charles T. Robbins[2,3] & Joanna L. Kelley [2,5]

Revealing the mechanisms underlying the reversible physiology of hibernation could have applications to both human and animal health as hibernation is often associated with disease-like states. The present study uses RNA-sequencing to reveal the tissue and seasonal transcriptional changes occurring in grizzly bears (*Ursus arctos horribilis*). Comparing hibernation to other seasons, bear adipose has a greater number of differentially expressed genes than liver and skeletal muscle. During hyperphagia, adipose has more than 900 differentially expressed genes compared to active season. Hibernation is characterized by reduced expression of genes associated with insulin signaling, muscle protein degradation, and urea production, and increased expression within muscle protein anabolic pathways. Across all three tissues we find a subset of shared differentially expressed genes, some of which are uncharacterized, that together may reflect a common regulatory mechanism. The identified gene families could be useful for developing novel therapeutics to treat human and animal diseases.

[1] Department of Integrative Physiology and Neuroscience, Washington State University, Pullman, WA 99164, USA. [2] School of Biological Sciences, Washington State University, Pullman, WA 99164, USA. [3] School of the Environment, Washington State University, Pullman, WA 99164, USA. [4] Department of Veterinary Clinical Sciences, College of Veterinary Medicine, Washington State University, Pullman, WA 99164, USA. [5] These authors contributed equally: Heiko T. Jansen, Shawn Trojahn, Michael W. Saxton, Joanna L. Kelley. Correspondence and requests for materials should be addressed to H.T.J. (email: heiko@wsu.edu) or to J.L.K. (email: joanna.l.kelley@wsu.edu)

Grizzly bears (*Ursus arctos horribilis*) have an annual cycle that includes hyperphagia and fat accumulation followed by winter-time hibernation in response to periods of food scarcity[1–4] (Fig. 1). Hibernation is characterized by lowered body temperature, inactivity, metabolic depression, and insulin resistance[5–8]. Yet, despite its annual occurrence, bears avoid the long-term detrimental effects that occur in obese, inactive, or fasting humans, such as bone loss, elevated blood nitrogen (azotemia), ketosis, hyperglycemia, and muscle protein catabolism and atrophy[7,9–13]. Thus, a series of carefully orchestrated and potentially unique mechanisms have evolved to maintain proper energy balance and metabolic health in bears throughout the year. Importantly, since these physiological changes in hibernators occur on an annual basis, mechanisms have evolved to naturally reverse these processes and thereby avoid any secondary complications[9].

Numerous studies of hibernating rodents have revealed tissue- and season-specific changes in gene expression[5,14,15]. Hibernating rodents and other small mammals share a common characteristic during hibernation, namely the ability to periodically arouse from deep torpor, which is an energetically expensive process[16,17]. Using patterns of expression in liver of arctic ground squirrels (*Urocitellus parryii*) compared to those of calorie-restricted, cold-exposed, and PPARα knockout mice, Xu et al.[18] revealed a signature of torpor and arousal. However, unlike the small-bodied hibernators, bears of the genus *Ursus* exhibit a continuous hibernation characterized by metabolic suppression and much smaller reduction in body temperature of 4–7 °C[2,19]. Therefore, by gaining a greater understanding of the cellular adaptations in different hibernating species one can envision new treatments being developed for human ailments[7,9,20].

Our understanding of the precise mechanisms of these reversible phenotypes in bears remains incomplete. Earlier studies in bears have identified some molecular changes in liver, heart, adipose, and skeletal muscle at different times of the year using targeted approaches such as microarray and PCR[21–23]. However, none have used unbiased methods, which could lead to the discovery of novel mediators. Thus, to provide a more complete picture of the molecular events involved, we performed RNA-sequencing (RNA-seq) on metabolically active tissues[24] obtained from the same six captive bears of both sexes across seasons, and in two bears over two consecutive hibernation seasons.

The unbiased methods employed in the present study reveal that hibernation is characterized by dynamic changes in gene expression. These changes are largely tissue specific; however, a subset of genes is DE in the same direction in all three tissues. The latter highlights the possible role of shared regulatory pathways and tissue crosstalk in hibernation. A more complete understanding of these changes during hibernation challenges the long-standing belief that hibernation is a static process and suggests a more nuanced regulation occurring on an on-demand basis. Our use of ribosomal-depleted RNA-seq allows us to develop a more holistic understanding of the role of long-non-coding RNAs. A greater knowledge of the reversible, and highly regulated, expression of the hibernation phenotype may facilitate the development of new treatments for human disease.

## Results

**Comprehensive RNA-sequencing of metabolic tissues in bears.** To determine the changes in gene expression across seasons in grizzly bears, we generated 218 Gigabases of ribosomal-depleted RNA-seq data for a total of 53 samples from three tissues in six individuals, which were sampled at three timepoints throughout 1 year with corresponding changes in body size and body temperature (Fig. 1a–c, Supplementary Data 1). The number of reads per sample after trimming for quality ranged from 24.1 to 59.9 million. The percent of reads mapped to the brown bear (*Ursus arctos*) reference genome (GenBank Accession: GCA_003584765.1, ref. [25]) ranged from 95.0% to 98.5%. The brown bear genome annotation set had 30,723 genes, of which 26,266 met our minimum expression cut-off (see Methods). The 26,266 genes were compared among the active (May), hyperphagia (late September), and hibernation (January) states in the same six individuals.

Two bears were sampled the following year in January (hibernation), which allowed us to assess the relative consistency

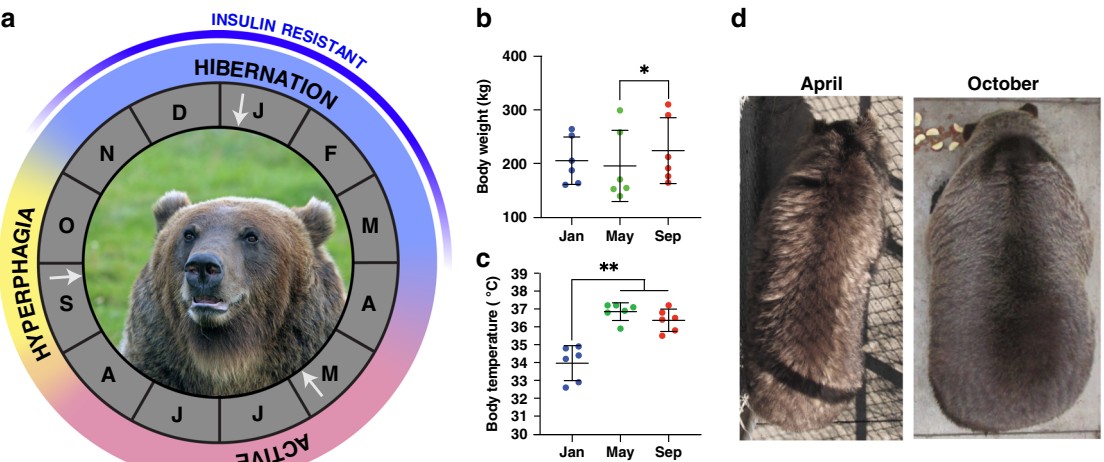

**Fig. 1** Major annual physiological cycles of grizzly (brown) bears. **a** Three major annual metabolic stages—hibernation, active, and hyperphagia—are indicated along with the letter abbreviation for the corresponding months of the year. Blue line along the perimeter indicates the approximate period of insulin resistance. Brown bear subcutaneous adipose, liver, and muscle samples were collected from six captive bears at the Washington State University Bear Research, Education, and Conservation Center during the metabolic stages. White arrows indicate sampling points. Photo credit to Donald Montemorra. **b** Body weight for each of the bears at the three sampling points indicated in **a**. *$p < 0.05$ (raw data in Supplementary Table 5). **c** Body temperature for each of the bears at the three sampling points indicated in **a**. **$p < 0.01$. $N = 6$ animals. Panels **b** and **c** include individual values, mean, and standard deviation (raw data in Supplementary Table 6). **d** Photograph of the same individual at the Washington State University Bear Research, Education, and Conservation Center coming out of hibernation (April) and entering hibernation (October)

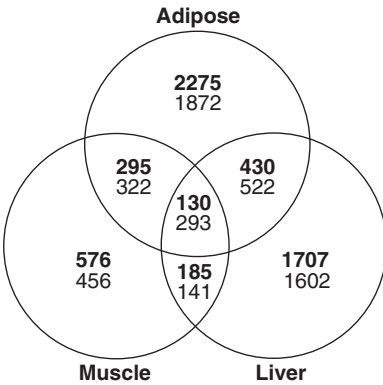

**Fig. 2** Differentially expressed genes in adipose, liver, and muscle at FDR < 0.05. Venn diagram depicting shared and unique variation in gene expression between hibernation and non-hibernation (average of active and hyperphagia) states across three tissues. Upregulated genes are in bold. N = 6 animals

of expression across years. When comparing samples taken during hibernation from the same individuals between years, we found that gene expression within tissues was highly correlated and often most correlated between samples taken during the same season (Supplementary Fig. 1).

**Hibernation results in widespread changes in gene expression.** Gene expression differed markedly among tissues (Supplementary Fig. 2) with striking differences in expression patterns evident between non-hibernating (active and hyperphagia) and hibernating states, especially in adipose and liver (Supplementary Fig. 3). Comparing the hibernating and non-hibernating states, 10,018 genes were differentially expressed (DE) in one or more tissues at a false discovery rate (FDR) <0.05 (Fig. 2, Supplementary Data 2). Adipose had the highest percent of DE genes (20.0% of all genes, 6139 genes) compared to liver (16.3%, 5010) and muscle (7.8%, 2398) (Fig. 2). A subset of genes was DE in the same direction in all three tissues, suggesting the possibility of shared regulatory mechanisms.

**Metabolic processes are overrepresented in hibernation.** In the entire dataset, we identified six modules of co-expressed genes with each module being significantly correlated with one tissue (Supplementary Fig. 4). As expected, the major component that explained patterns of co-expression of genes was differences among tissues. Therefore, we split the dataset by tissue and in each tissue we identified modules of co-expressed genes which resulted in 23 modules for adipose, 13 modules for liver, and 30 modules for muscle and one set of unassigned genes (gray) per tissue (Supplementary Data 6). Gene ontology (GO) enrichment analysis for each module revealed a large number of metabolic process related terms (Supplementary Tables 1–3). There were several modules of co-expressed genes that were significantly correlated with hibernation (Supplementary Fig. 5). For each tissue we also calculated the correlation between each module and other seasons and sex (Supplementary Fig. 5). In the module of co-expressed genes most highly correlated with hibernation in each tissue (Supplementary Fig. 6), none of the 30 most connected genes (hubs) were shared among tissues. These genes included six long non-coding RNAs in adipose, two in liver, and one in muscle.

**Gene expression during hyperphagia is tissue specific.** Hyperphagia was reflected in limited changes in gene expression in liver and muscle when compared to active season. Specifically, no

genes were DE between hyperphagia and active season in muscle and only three were DE in liver (*IGFALS*, *FADS2*, *GPT*). In contrast, 984 genes were DE in adipose, consistent with a selective increase in adiposity occurring in the autumn. Of those 984 genes, 267 were upregulated and 717 were downregulated in hyperphagia compared to active season (Supplementary Data 7). Hyperphagia clearly reflected a transition phase in adipose as many of the DE genes had intermediate expression, trending down or up between active and hibernation states (Fig. 3).

**Different bear species exhibit similar changes in expression.** A subset of genes in the current study were found previously to be DE in a microarray study in black bears (*Ursus americanus*)[22], with the discrepancies possibly reflecting subtle species differences. That analysis identified 319 genes in liver that were DE. Of the 319 genes, 285 had a direct match in our dataset and 78.9% (225 genes) of the transcripts shared among studies changed in the same direction (Supplementary Fig. 7). Of the complete set of DE genes overlapping between studies (285 genes), 164 genes were also significantly DE (FDR, *p* < 0.05) in our study, with 87% having a log fold change in the same direction.

## Discussion

Our study identified sweeping changes in gene expression within the same animals sampled at different times of year. The most predominant changes occurring in hibernation were in metabolism-related genes and pathways in adipose tissue, as demonstrated by the differential gene expression (Supplementary Data 4) and gene set enrichment (Supplementary Data 5) analyses. These results are consistent with the necessary reduction in energy expenditure during hibernation[26]. In contrast to the large number of DE genes we found in adipose in bears, recent studies in captive dwarf lemurs and elephant seal pups found fewer than 100 DE genes between physiological states similar to the active (feeding) and hibernation (torpid or fasting) seasons[27,28]. In lemurs, that number increased to 377 DE genes when comparing fattening (similar to bear hyperphagia) and torpid states[29]. However, even when down-sampling our data to match sample sizes within these studies there were dramatic differences in the number of DE genes between the species (lemur 337 DE genes of 10,745 total vs. bears with 3446 DE genes of 17,564 total). Although the reason for the large differences in gene expression responses between species is unknown, its large magnitude suggests highly diverse genetic bases for maintaining energy balance while fasting.

The complexity of the hibernation phenotype in bears was underscored by the number of genes in modules of co-expressed genes correlated with hibernation (Supplementary Tables 1–3). Moreover, hubs of those modules (i.e. the most highly connected genes) contained multiple long non-coding RNAs (Supplementary Fig. 6). Indeed, the gene with the highest connectivity in adipose is a long non-coding RNA (*LOC113264107*), thus, highlighting the role of non-coding RNAs in hibernation.

Hibernating bears exhibit a dramatic reduction (up to 75%) in metabolic rate[19,30] but still require energy in the form of ATP for the maintenance of bodily functions. Metabolic fuels—glucose, fatty acids, and ketones represent the primary sources for ATP generation. Under conditions of fasting or starvation all three sources are utilized to varying extents (see ref. [31] for review). In adipose and liver of hibernating grizzly bears we found that glucose breakdown via glycolysis is greatly reduced based on reductions in hexokinase expression (*HK1* and *GCK*). These findings are similar to those made in Asiatic black bears[32].

During fasting, gluconeogenesis is expected to increase in the liver, the primary organ for glucose production. In support of

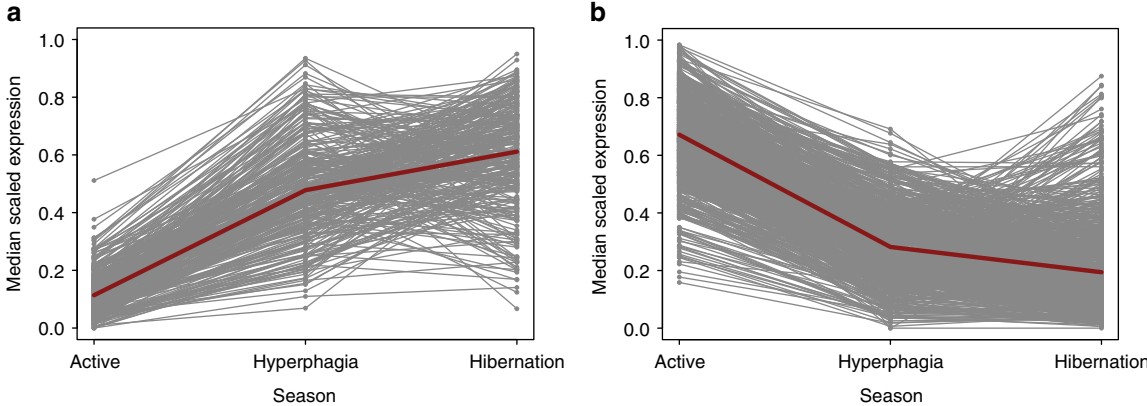

**Fig. 3** Seasonal patterns of gene expression in adipose. **a** Median scaled expression for genes that are significantly upregulated (Supplementary Data 7) in hyperphagia compared to active season; median values are connected with a red line. **b** Median scaled expression for genes that are downregulated in hyperphagia compared to active season; median values are connected with a red line

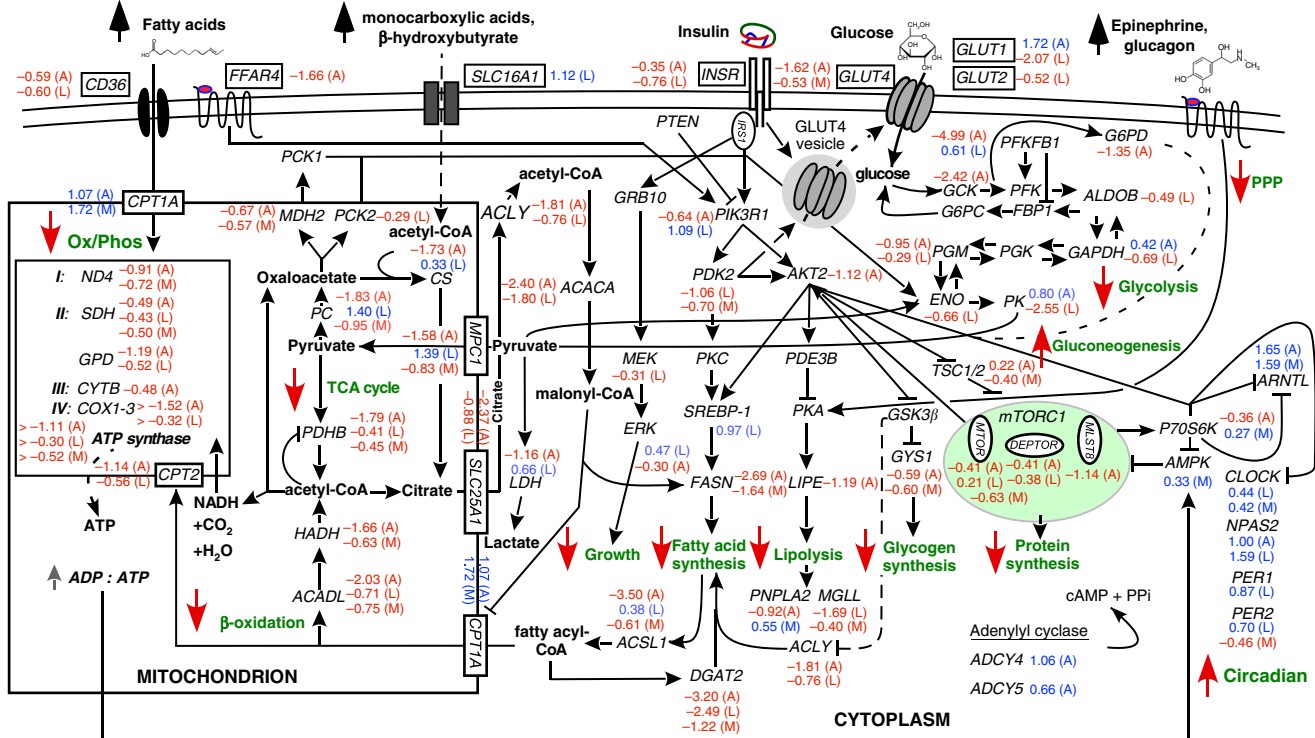

**Fig. 4** Intermediary metabolism is suppressed in the fasted state. Schematic illustrating the regulation and endpoint of major metabolic pathways in adipose (A), liver (L), and muscle (M). $\text{Log}_2$ fold change is indicated next to each gene; red numbers indicate significantly decreased expression in hibernation, whereas blue indicates significantly increased expression in hibernation ($N = 6$ animals, FDR < 0.05; Supplementary Data 2)

this, conversion of pyruvate to oxaloacetate (OAA), the first step of gluconeogenesis and mediated by pyruvate carboxylase (PC), was greatly increased during hibernation (Fig. 4)—a finding similar to that made previously in black bears[22,32]. Additionally, while expression of the key regulatory enzyme, fructose 1,6-bisphosphatase 1 (*FBP1*), was unaffected in hibernation, unlike in black bears[22], the expression of fructose-2,6-bisphosphatase 1 (*PFKFB1*) was significantly increased (Supplementary Data 2), thus favoring gluconeogenesis over glycolysis. The final step in glucose production, mediated by conversion of glucose 6-phosphate to glucose by glucose 6-phosphatase (*G6PC*) was unaffected, which is similar to observations in black bears[32] but opposite to that of hibernating Arctic ground squirrels[33]. Lastly, previous reports of increased glucagon[8,34], serum lactate[35], and glycerol (described below) during hibernation are all consistent

with a net increase in gluconeogenesis at this time of the year in bears.

Glucose uptake by tissues via insulin is tightly regulated. Although bears do not become overtly hyperglycemic, bears and other hibernators exhibit annual periods of insulin resistance[7,8,36]. This reversible process makes hibernation an excellent biological system to study the etiology of insulin resistance[7]. Fasting itself can be associated with insulin resistance[37]; however, withholding food during the active season in bears does not replicate the hibernation phenotype[38]. Conversely, feeding bears during hibernation only partially restores insulin sensitivity[36]. These findings highlight the seasonal rhythm of insulin sensitivity in bears. Within the insulin pathway key genes were downregulated during hibernation (Fig. 5, Supplementary Data 2). By contrast, two genes that reduce recruitment of *PDK1*

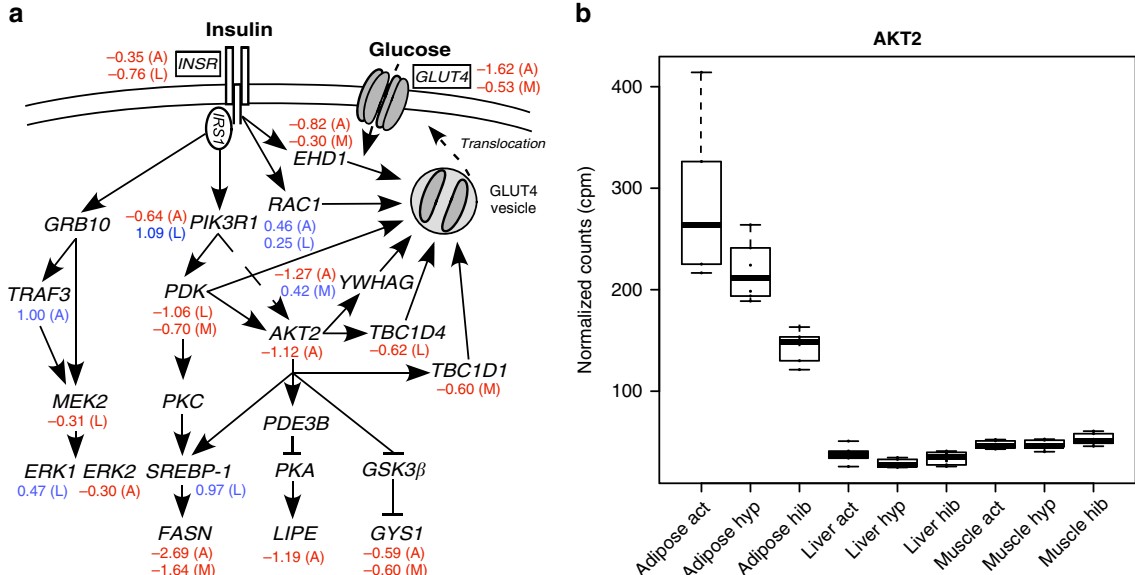

**Fig. 5** Transcriptional regulation of key insulin-signaling steps. Schematic illustrating the regulation and endpoint of a major metabolic pathway. Log$_2$ fold change is indicated next to each gene; red numbers indicate significantly decreased expression in hibernation, whereas blue indicates significantly increased expression ($N = 6$ animals, FDR < 0.05, Supplementary Data 2). **a** Major components of the insulin-signaling pathway illustrating changes occurring in tissues (A—adipose, L—liver, M—muscle) between hibernating and non-hibernating states. Dashed line indicates vesicle translocation. **b** Seasonal changes in *AKT2* expression per tissue and season (active—Act; hyperphagia—Hyp; hibernation—Hib). All box plots have the following elements: center line, median; box limits, upper and lower quartiles; whiskers, 1.5× interquartile range; points, outliers

and *AKT2* to the insulin signaling pathway (*SHIP1* and *SHIP2*) were upregulated in adipose tissue in hibernation. Collectively, these DE genes would impact the penultimate step of insulin-dependent glucose uptake via reduction in translocation of vesicles containing the primary glucose transporter mediating insulin's metabolic actions (GLUT4)[39] to the phospholipid bilayer. This then would lead to a reduction in insulin mediated glucose uptake, as is observed in hibernation[8]. Indeed, we observed a significant reduction in *GLUT4* expression in both adipose and muscle (Supplementary Data 2). These results highlight adipose and muscle as sources of insulin resistance during hibernation in bears[8]. In humans, genome-wide association studies (GWAS) have identified many genes associated with insulin resistance, and 16 were found to be clinically responsive to treatment[40,41]. Among these, 14 were DE in our bears between hibernating and non-hibernating states (Supplementary Table 4), indicating possible conserved pathways regulating insulin sensitivity between species.

Liver, another primary insulin responsive tissue, had unique shifts in gene expression in hibernation. While there were large changes in expression across the hepatic transcriptome (Supplementary Fig. 3, Supplementary Data 2), no pattern within the insulin-signaling pathway matched what is seen in adipose or muscle. The insulin receptor (*INSR*) was significantly down-regulated, as is seen in adipose, but most downstream changes in signaling that lead to reduced vesicular translocation were absent in liver. However, there is still likely a reduction in hepatic glucose uptake due to a downregulation of the primary hepatic glucose transporters, *GLUT1* and *GLUT2* (ref. [42]). This points to tissue-specific responses within the insulin-signaling pathway that lead to the observed global insulin resistance.

A reduction in tissue glycolysis and thus a reduction in ATP production during hibernation could be offset by the use of other fuel sources, such as fatty acids. These are stored in abundance in the fat droplets of (mostly) adipocytes in mammals and are especially important for hibernators as fatty acids represent the most abundant fuel source during fasting. Serum concentrations

of fatty acids are increased during hibernation whereas the expression of genes involved in fatty acid synthesis are greatly reduced[43–46]. Consistent with this, we found fatty acid synthase (*FASN*) was reduced in adipose and muscle during hibernation, as was expression of the rate-limiting enzyme in long-chain fatty acid synthesis, acetyl-CoA carboxylase (*ACACA*) in both adipose and liver (Fig. 4). Thus, enhanced production of lipids during the active season for fuel use during hibernation is entirely consistent with the expected shift from glucose to lipid usage[47,48]. Based on this, one would also expect the activity of tissue lipases to be increased during hibernation, thereby liberating free fatty acids and glycerol. While this was confirmed in Asiatic black bears[23] and marmots[49], we saw only *reduced* expression of the major lipolytic genes, *LIPE* (hormone sensitive lipase, also known as HSL), *ATGL* (adipose triglyceride lipase), and *MGLL* (mono-glyceride lipase) (Supplementary Data 2). However, it is equally plausible, and less energetically costly, that the expression of a negative regulator of lipase activity was reduced, thereby indirectly activating lipolysis. Indeed, we found that the expression of the negative regulator lipoprotein lipase (*LPL*)[50] was greatly suppressed in adipose, a finding similar to that in marmots[49].

We found increased expression of *CPT1a* and *b* (carnitine O-palmitoyl transferase 1), which are involved in mitochondrial beta oxidation in adipose during hibernation, consistent with the findings of Shimozuru et al.[23]. Thus, it would seem that the transport of fatty acids into mitochondria is favored over glycerol production and fatty acid release from the cell. This would be somewhat surprising given the elevated circulating levels of fatty acids and glycerol that have been observed in hibernating bears previously[8,23,35,43]. Nevertheless, it should be cautioned that changes in enzyme activity and exogenous stimuli (e.g., adrenergic stimulation of lipolysis) occurring on an as-needed basis cannot be ruled out. In muscle, fatty acid catabolic genes tended to be overexpressed, somewhat consistent with previous findings in Arctic[33] and 13-lined ground squirrels[51].

Beyond metabolism, hibernating bears provide insight into disuse muscle atrophy as they lose considerably less muscle mass

than what is predicted for humans over a similar period of inactivity[52]. Increased protein synthesis within muscle has been proposed as an explanation for the lack of muscle loss[53]. Preferential induction of protein synthesis during hibernation was supported by our analyses, similar to previous studies[21,22]. In our study, while the translation GO category (GO: 0006412) is overrepresented in both the set of upregulated and downregulated genes in muscle, the GSEA result based on log$_2$-fold change ranking was consistent with other studies (Supplementary Fig. 8). Liver also showed an enrichment for protein synthesis whereas adipose shows the opposite signal with a depletion of translation related genes expressed during hibernation. The timing of hibernation sample collection in March in the earlier study[21], which could potentially be a more transitional phase between hibernation and the active season, as compared to the January sample collection in this study during mid-hibernation suggests that protein synthesis is important throughout hibernation in muscle and liver.

We observed downregulation of important genes within the autophagy–lysosome and ubiquitin–proteasome pathways, perhaps further explaining the lack of muscle atrophy observed in hibernating bears. For the autophagy–lysosome pathway, *MLP3A (LC3)* is the key gene underlying autophagosome formation and it is downregulated in bears, potentially inhibiting this pathway. The ubiquitin–proteasome pathway appears to be regulated via a reduction in the expression of transfer protein ZFAND5. Surprisingly, there was not a significant downregulation of *ATROGIN1/MAFBX* or *MURF1*. However, there was a downregulation of *ASB2*, an E3 ubiquitin ligase with high specificity for filamin B found within the Z-line[54]. These observed changes in gene expression make *MLP3A*, *ZFAND5*, and *ASB2* ideal targets for future research on muscle atrophy.

Bears may also provide useful information for developing novel treatments for kidney diseases as they do not urinate or defecate during hibernation and thus likely have evolved unique mechanisms to handle waste products. A previous study hypothesized that protein degradation increases during hibernation, amino acids are cycled through the urea cycle, and ammonia is reabsorbed through the gut lumen[38]. This would result in the creation of new amino acids for protein anabolism and gluconeogenesis. Elevated circulating levels of the gluconeogenic amino acid alanine in bears[9] could provide an additional substrate for glucose production while at the same time increasing urea concentrations. However, as our results reveal a significant downregulation of the genes of both the urea pathway (Fig. 6) and important transaminases for metabolism of amino acids (*ALAT1* and *ALAT2*), we conclude that urea recycling plays only a minor role in the hibernation phenotype. In support of this, we found a significant decrease in *GPT* in both liver and muscle (Supplementary Data 2), similar to findings in black bears[32], indicating that new alanine is not being formed by the recycling of urea. Thus, amino acids are most likely preserved through a downregulation of proteolytic enzymes in both the autophagy–lysosome and ubiquitone–proteosome pathways described above.

Gene expression in muscle of hibernating grizzly bears differed considerably from that in ground squirrels during torpor or interbout arousals[51]. For example, we found very little change in genes involved in muscle ß-oxidation (e.g., *ACADS*, *ACADVL*, *ACAD10*, *HADHA*, etc.). Only *ECH1* was similar in bears and squirrels in that it was upregulated in hibernation. Additional disparities were observed for genes involved in fatty acid metabolism (e.g. *FABP3*, *ACOT1*, *ACSL*), protein synthesis (e.g. *AKT1*, *MTOR*), IGF signaling (e.g., *IGF1,2*), and protein degradation (e.g., *FOXO1,3*, *ZFAND5*). In adipose (white adipose), the similarities between bears and ground squirrels were limited to *FABP4* (no seasonal changes) and up-regulation of *HMCS2*[55]. An

increase in albumin (*ALB*) expression was observed in squirrel adipose[55]. Although we did not find this in bear adipose, liver expression of *ALB* was significantly increased during hibernation (Supplementary Data 2), consistent with previous observations in brown bears[9]. Thus, the major carrier of serum lipids to tissues, elevated albumin could provide increased fuel to tissues even without stimulating lipolysis. It appears that different strategies have evolved in bears for lipid metabolism and protein synthesis and breakdown during hibernation. This difference is further illustrated by the failure of bears to develop hyperketonemia or ketoacidosis during hibernation compared to squirrels and fasting humans[56,57].

Whether a common hibernation signature exists between these different species is still unclear. Nevertheless, by gaining a greater understanding of the cellular adaptations, both unique and shared, by different hibernating species one can envision new treatments being developed for human ailments, such as obesity, type 2 diabetes, atherosclerosis, and chronic kidney disease[7,9,20]. However, it is important to note that gene expression does not necessarily reflect protein levels and fails to capture phospho-proteomic changes which may have substantial impact on a given pathway. While expression is useful for identifying possible regulatory mechanisms (i.e., long non-coding RNAs), future proteomic and enzymatic functional studies will be necessary to develop a more holistic understanding of hibernation.

Based on the extensive changes in tissue expression profiles that occur between active, hyperphagia, and hibernation states in grizzly bears, our results suggest a far more complex regulation of cellular function. Clearly, a vast and tissue-specific network of genes underlies the distinct seasonal and reversible phenotype of bears. A deeper understanding of the intricacies involved in these seasonal changes could provide a platform from which to explore reversible adaptations. This information could have direct implications for the treatment of metabolic diseases, the discovery of potentially novel mediators of metabolic homeostasis, and treatment of a variety of muscle wasting pathologies.

## Methods

**Animals and facilities**. Captive grizzly bears were housed at the Washington State University Bear Research, Education, and Conservation Center in Pullman, WA. Bears were active from mid-March to early November. During the active season bears were fed a base diet consisting of Hill's Science Diet® dry adult dog food (21% protein, 12.5% fat, 3% fiber, 10% moisture, 400 IU/kg vitamin E, 85 mg/kg vitamin C, 2.5% omega-6 fatty acids; energy content 3659 kcal ME/kg; Hill's Pet Nutrition, Topeka, KS) supplemented daily with fresh fruit and meat or fish. During this period all bears had access to a 0.56 ha irrigated, outdoor enclosure containing abundant forbs and grasses and both indoor pens (3m × 3 m × 2.5 m) and outdoor runs (3 m × 3 m × 5 m). In October, food availability was reduced and then discontinued by early November. Bears hibernated in their home dens with access to runs, water ad libitum, and straw bedding. Bears were exposed to natural temperature and light conditions throughout the year. Animals were maintained according to the American Society of Mammalogists' guidelines[58] and the Bear Care and Colony Health Standard Operating Procedures. All procedures were approved by the Washington State University Institutional Animal Care and Use Committee. Specific details regarding housing, diet, and care of bears can be found in ref. [8].

**Sample collection**. Samples were collected from six bears (2 females, 4 males, ranging from 5 to 13 years old) in May 2015, late September 2015, and January 2016, to represent the summer active, autumn hyperphagic, and winter hibernation periods, respectively. Muscle, liver, and adipose tissue samples were collected for a total of 54 samples. Bears were first anesthetized using the protocol described in ref. [59]. Subcutaneous adipose samples were collected using a 6 mm punch biopsy (Miltex, York, PA, USA) as described in ref. [8], while skeletal muscle (gastrocnemius) and liver tissue samples were collected with a 14G tru-cut biopsy needle (Progressive Medical International, Vista, CA, USA). Samples were immediately flash frozen in liquid nitrogen and transferred to a −80 °C freezer, where they were stored until RNA extraction. All samples were collected between 0800 and 1200 hours. Procedures for all experiments were approved by the Institutional Animal Care and Use Committee at Washington State University (Protocol #04922).

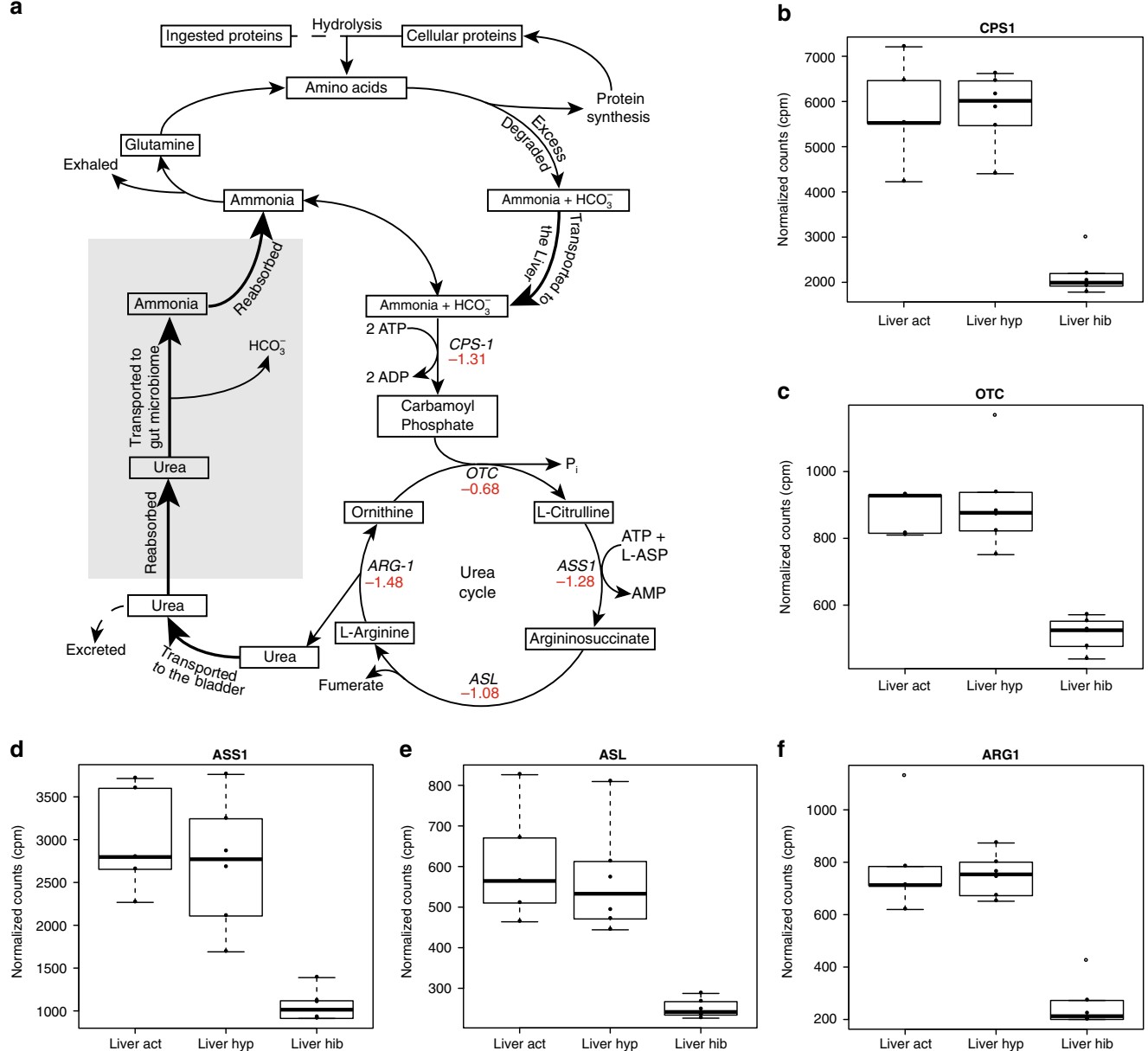

**Fig. 6** Urea cycle downregulated during hibernation. **a** Schematic of the path proteins (either ingested or recycled) take in the body. Thick arrows indicate movement of compounds through the body. Numbers under enzyme names indicate log$_2$ fold change observed during hibernation in liver. Red numbers indicate significantly decreased expression in hibernation (FDR < 0.05, Supplementary Data 2). Dashed lines show pathways that occur normally during active season but do not occur during hibernation. The gray box highlights the step of urea reabsorption and transport to the gut microbiome with produced ammonia being absorbed across the gut lumen. **b–f** Normalized counts (counts per million, cpm) from liver for the five genes in the urea cycle plotted across active (Act), hyperphagia (Hyp), and hibernation (Hib). All box plots have the following elements: center line, median; box limits, upper and lower quartiles; whiskers, ×1.5 interquartile range; points, outliers open circles. $N = 6$ animals

**RNA extraction**. Disruption and homogenization for all samples was completed with a TissueLyser LT (Qiagen, Redwood City, CA, USA) and extractions were performed using the RNeasy fibrous tissue mini kit (Qiagen, Valencia, CA, USA) for muscle tissue and the RNeasy mini kit (Qiagen) for liver and adipose tissue. Automation of several steps in each extraction protocol were executed by a QIA-cube (Qiagen). Liver and adipose tissue extractions differed from the manufacturer's protocol in that the RLT lysis buffer was replaced with QIAzol (Qiagen) to aid in the removal of proteins and lipids prior to loading the sample on a column. Therefore, up to 25 mg of frozen tissue was placed immediately into a 2 mL tube containing 600 µL QIAzol and a stainless-steel bead. After tissue homogenization 120 µL chloroform was added to the solution and vortexed for 15 s. The samples were then incubated at room temperature for 1 min prior to a 15-min centrifugation at 4 °C and 14,000 r.p.m. Finally, 350 µL of the aqueous phase was removed and placed into a new 2 mL tube for loading into the QIAcube. RNA yield and quality were assessed using a Qubit 2.0 (Invitrogen, Carlsbad, CA, USA) and Bioanalyzer 2100 (Agilent Technologies, Santa Clara, CA, USA),

respectively. The amount of total RNA per milligram of tissue varied by season (Supplementary Fig. 9). After quality control, the remaining RNA was frozen at −80 °C. One liver sample failed RNA extraction.

**Library preparation and sequencing**. RNA-sequencing libraries were prepared using the TruSeq Stranded Total RNA Prep with Ribo-Zero Gold kit according to the manufacturer's instructions (Part #15031048 Rev.E; Illumina, San Diego, CA, USA) with one library per sample. Ten rounds of amplification were used for library enrichment. To assess the quality and determine the molarity, each library was run on an Agilent Bioanalyzer 2100. The 53 samples, which were individually barcoded during library preparation, were then pooled in equimolar concentrations using the Bioanalyzer data. The pool was initially run on an Illumina MiSeq to ensure that each library was present in an equal proportion. The MiSeq data were then used to identify which libraries were under-represented in the pool. An additional volume of each of the under-represented libraries was then added to the

original pool to balance their representation in the final pool. The final adjusted pool was then sequenced on seven lanes of an Illumina HiSeq 2500 System (v4 reagents, paired-end, 100 basepair (bp) reads) at the Washington State University Genomics Core in Spokane, WA.

**Mapping**. Raw sequencing reads were assessed with FastQC (version 0.11.4)[60]. Reads were subsequently trimmed using Trim Galore! (run with Cutadapt version 0.4.2)[61] with a minimum quality score of 20 and 10 bp was removed from the 5′ end of all reads. Only paired reads, each at least 50 bp in length, were retained for the analysis. Trimmed reads were mapped to the brown bear (*Ursus arctos*) reference genome assembly (GCF_003584765.1)[25]; we also mapped the reads to the polar bear (*Ursus maritimus*) reference genome assembly (GCF_000687225.1)[62] using HISAT2 (version 2.1.0)[63] and results were consistent. Mate pair information was verified and fixed where necessary using Picard tools (version 2.2.1)[64]. Output SAM files were converted to BAM format and sorted by coordinates with SAMtools (version 1.2)[65]. Mapping statistics were assessed using Picard tools[64]. After mapping, one sample (CAA) was removed from analyses because the expression analysis revealed that a hair follicle was inadvertently sampled with the adipose tissue.

**Gene counts matrix**. We used StringTie (version 1.3.4d) with the -eB flags to estimate the number of reads mapping to each gene in the reference annotation set of the brown bear genome (NCBI *Ursus arctos horribilis* Annotation Release 100). To generate a gene counts matrix, we ran prepDE.py (a Python script provided with StringTie) on the StringTie output. For all analyses we removed annotated ribosomal RNA genes (nuclear and mitochondrial) since we performed an RNA-sequencing library preparation method with a ribosomal subtraction step.

**Gene annotation**. To annotate the genes in the brown bear reference set (NCBI *Ursus arctos horribilis* Annotation Release 100), we extracted the longest coding transcript for each gene using gffread[66]. All coding genes were annotated using a BLASTx (version 2.2.31)[67] search against the human SwissProt database (critical *E*-value: 0.00001; access date 6/9/2018)[68]. For all BLAST searches, we retained the top BLAST hit per gene based on the top high-scoring segment pair. A total of 19,967 were annotated as protein coding, 19,621 of those genes resulted in a significant BLAST hit (*E*-value < 0.00001) in the human SWISSPROT protein database. The BLASTx results were then imported into Blast2GO (version 5.2.0)[69] to explore the putative biological functions of candidate gene sets. We annotated sequences with a match in the SwissProt database with GO IDs[70] for the gene ontology enrichment analysis. We tested for the enrichment of specific GO IDs in candidate genes relative to the entire set of genes using a one-tailed Fisher's Exact Test with an FDR < 0.05 in Blast2GO.

**Large-scale expression patterns**. Multidimensional scaling (MDS) was used to visualize the overall variation in gene expression as well as tissue-specific patterns. To visualize overall variation in gene expression for all samples and tissue-specific patterns, MDS and sample clustering was queried. Genes with very low expression (less than 0.5 counts per million in at least three samples) were removed from further analyses. The data were first transformed using the variance-stabilizing transformation function in the DESEQ2 (version 1.22.2) package in R (version 3.5.2)[71], which transforms data normalized for library size to a log$_2$-scale[72]. Normalized read counts from the top 10,000 variable genes were analyzed with MDS first across all samples (Supplementary Fig. 2), and then within each tissue separately using *plotMDS* in the limma package (version 3.38.3)[73] in R (Supplementary Fig. 3). We also created a hierarchical clustering dendrogram for each tissue based on the Euclidian distance between samples using *hclust* and *dist* in R (Supplementary Fig. 3).

**Comparison across years**. RNA-sequencing libraries were prepared using the TruSeq Stranded Total RNA Prep with Ribo-Zero Gold kit according to the manufacturer's instructions (Part #15031048 Rev.E; Illumina, San Diego, CA, USA). The raw sequencing reads were trimmed, mapped, and verified using the same protocol as above. All samples were subsampled to 5.6 million reads using seqtk (version 1.3-r106; https://github.com/lh3/seqtk) to ensure varying numbers of reads did not influence the correlation between years. A gene counts matrix was generated using prepDE.py on the StringTie output using all samples. Genes with very low expression (<0.5 counts per million in at least three samples) were removed from further analyses. These data were then transformed using the variance-stabilizing transformation function in the DESEQ2 package in R[71]. The normalized gene counts for each tissue from bear one and bear two were plotted against each other and the adjusted $R^2$ was calculated in R.

**Differential expression**. Differential expression was quantified based on normalized read counts using the Bioconductor package edgeR (version 3.24.3)[74,75] in R. Our study had multiple factors (individual and season); therefore, for each analysis we implemented a generalized linear model (GLM) in the edgeR package. To test for statistical significance, we used the GLM quasi-likelihood *F* test

(function *glmQLFTest* in edgeR) with Benjamini–Hochberg correction and FDR < 0.05. To compare the hibernation and non-hibernating states, we compared expression levels between the hibernation samples and the average of active (summer) and hyperphagia (autumn). An additional comparison was made to identify genes that were DE between active and hyperphagia seasons. The Gene Ontology analysis was performed with Blast2GO using a one-sided Fisher's exact test with an FDR < 0.05.

**Gene set enrichment analysis**. Gene set enrichment analysis was performed in GSEA (version 3.0). Genes were ranked according to the correlation with hibernation based on the log$_2$ fold change between hibernation and non-hibernation samples. The gene sets were the canonical pathways (CP) from the curated gene set (C2) and the GO biological process (BP) from the GO gene sets (C5). The enrichment score (ES) was calculated within GSEA and reflects whether the gene set is overrepresented at the top or bottom of the ranked list. The enrichment was determined using 1000 permutations.

**Weighted gene co-expression networks**. To identify modules of co-expressed genes, we split the samples by tissue type to evaluate each tissue separately. We conducted a weighted gene co-expression network analysis (WGCNA) on all expressed genes meeting the expression cut-off using the R package WGCNA (version 1.66)[76], as described in ref. [77]. Before constructing the networks, we determined a soft thresholding power of 9, 6, and 8 for adipose, liver, and muscle data, respectively, which was identified for each tissue using *pickSoftThreshold* in the WGCNA package in R, such that the model fitting index $R^2 > 0.9$. To identify signed networks the *blockwiseModules* function in the WGCNA package in R was used with networkType = signed, TOMType = signed, minModuleSize = 30, reassignThreshold = 0, mergeCutHeight = 0.25, maxBlockSize = 25000, and all other default options.

**Statistics and reproducibility**. Details of each statistical method can be found above (Methods) and in the code at https://github.com/jokelley/brownbear-rnaseq-tissue-act-hyp-hib. Sample size was defined as the number of bears ($N = 6$), each serving as an independent biological replicate. Three tissues from each bear were sampled once at each sampling point (Fig. 1a). Technical replicates were not used. An experimental replicate included an independent analysis of gene expression in adipose from two consecutive hibernation seasons ($N = 2$).

**Ethics approval**. All protocols using grizzly bears (*Ursus arctos horribilis*) were approved by the Washington State University Institutional Animal Care and Use Committee (IACUC). Protocol #'s 03875 and 04922. Males and females were included (ages ranged from 4 to 10 years).

**Reporting summary**. Further information on research design is available in the Nature Research Reporting Summary linked to this article.

## Data availability
All data are available in Genbank BioProject PRJNA413091. Summaries of genes, GO terms, GSEA, and differentially expressed genes can be found in Supplementary Data 1–7. All other data are available from the corresponding authors on request.

## Code availability
All code is available at https://github.com/jokelley/brownbear-rnaseq-tissue-act-hyp-hib

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

## Acknowledgements

We appreciate funding and support provided by the Interagency Grizzly Bear Committee, USDA National Institute of Food and Agriculture (Hatch project WNP00226), International Association for Bear Research and Management, T. N. Tollefson and Mazuri Exotic Animal Nutrition, and the Raili Korkka Brown Bear Endowment, Nutritional Ecology Endowment, and Bear Research and Conservation Endowment at Washington State University and a National Science Foundation Graduate Research Fellowship (to S.T., 1347973). We appreciate the feedback from the Cornejo and Kelley labs on this manuscript, and particularly the assistance of Scott Hotaling. J.L.K. would like to acknowledge her writing group. This research used resources from the Center for Institutional Research Computing at Washington State University.

## Authors contributions

H.T.J., M.W.S. and J.L.K. conceived this study. H.T.J., S.T., M.W.S., C.R.Q., O.E.C., B.D. E.H, O.L.N, C.T.R and J.L.K. performed experiments and contributed to data analysis. H.T.J., S.T., M.S. and J.L.K. wrote the manuscript.

## Additional information

**Competing interests:** The authors declare no competing interests.

