## [Peer Review File · Communications Biology]

Reviewers' comments:

Reviewer #1 (Remarks to the Author):

The manuscript by Jansen et al. uses RNAseq to examine changes in gene expression patterns between summer active bears, summer active hyperphagic bears, and hibernating bears. The strength of the manuscript is its use of non-biased methods to provide a more complete picture of the molecular events that underlie the hibernation phenotype (it's also important that they have measured at least a couple of summer states [sampling on post-hibernation bears that have increased metabolism but not yet fed would have been illuminating and it's unfortunate this group was not included]). Additionally, because the sample design involves repeated measures, individual variability is accounted for. The manuscript links changes in gene expression to medical disorders (with a focus on insulin-resistance), which should provide broad appeal. The most interesting question in terms of insulin resistance in bears is how the process is reversible, but unfortunately this is beyond the scope of the current study. Many of this study's findings align with prior studies of gene expression during hibernation in bears (citations 5 & 6 in the manuscript), though the design of the current study is more elegant and the number of genes investigated is much larger due to the newer approaches employed. Overall, this manuscript has a similar issue as many other the hibernation genomics papers – the results of gene expression studies are contributing very detailed knowledge regarding processes that have long been known (e.g., that hibernators exhibit insulin resistance – reviewed by Martin 2008 *Diabetes and Vascular Disease Research*, 5, 76-8) but the question is how does this genomic data move us closer to using this animal model to solve this underlying medical issue) – this is the key element that seems to be lacking here – some form of a path forward, rather than simply a much more detailed understanding of the molecular mechanisms involved in the model organism.

Specific comments:

Line 64: Instead of "small subset" – just state this was done in two animals.

Line 84: "active plus hyperphagia" is somewhat confusing

Lines 148-152: This comparison is a little too simple – you are comparing 3 times as many genes in the bear study compared to the Lemur study and sample size is 6 per group in bears vs. 4 per group in lemurs – thus, some of the differences in # of "significantly" DE genes may simply reflect differences in power between experiments. I would think there are more quantitative ways to make this comparison.

Line 174-176: "Some genes in the insulin metabolic pathway also have multiple paralogs. Of these gene families, including PIK3, PP2A, and PKC, we found variability in differential expression of paralogs in adipose." This is unclear – restate this in another way

Line 186: Don't use the term "starvation" – replace with "fasting" throughout the manuscript

Lines 189: "Together, the results in hibernating bears add support to the notion that insulin resistance in adipose plays an important role in energy homeostasis and metabolic disease." I find this sentence confusing – how does the hibernating bear results inform about the role of adipose in metabolic disease? The hope might be to provide insight into how insulin sensitivity is reversible, but it's hard to say what underlies this reversible phenotype based off of gene expression data.

Lines 287-290: "Based on the extensive changes in tissue expression profiles that occur between active, hyperphagia and hibernation states in grizzly bears, our results suggest a far more complex regulation of cellular function than could be induced by a fast acting "hibernation induction trigger" sought by previous investigators" citation 41 - This seems a bizarre (and old – 1987) citation to use that fails to describe the current view of hibernation research. I agree entirely with the sentiment – but I think this sentiment is shared among the majority of hibernation researchers and seems off-target with the scope of most of the manuscript.

Supplemental Fig 1 – This isn't really a test of how representative the tissue is for that state – at least some comparison's between other states would be helpful (i.e., I expect expression may be highly

correlated within tissues, even across states). I don't doubt that it is representative, but I think there are better ways to illustrate this. Supplemental Fig 2 is more informative in this regard (could simply show the different years with different shapes).

Supplemental Fig 3 – there is no explanation of the different shapes (square vs circle) in the leftmost panels.

Supplementary Fig 6 – I assume that black dots were found to be differentially expressed by Fedorov et al. but not in the current study? Are there also genes found to be DE in the present study that were not DE in Fedorov et al?

Reviewer #2 (Remarks to the Author):

Communications Biology: Manuscript COMMSBIO-19-0479

Hibernation induces widespread transcriptional remodeling in metabolic tissues of the grizzly bear
Jansen et al.

SYNOPSIS

This study performed RNA-sequencing of adipose, liver, and skeletal muscle of the brown bear. Gene expression was compared among the active, hyperphagia, and hibernation states, extensive changes being shown to occur during hibernation. By using extensive bioinformatics, the authors describe global differences between tissues and/or seasons. Then they focus on insulin signaling, showing that their data would be in line with insulin resistance during hibernation. Changes related to the intermediary metabolism and protein balance are also briefly discussed and appear to be in line with muscle preservation in hibernating animals. The authors conclude that their results may influence positively human health in relation with metabolism, insulin sensitivity, and muscle atrophy.

General comments:

The work appears to have been done carefully and the article presents unique datasets that extend far beyond existing ones. Congrats to the authors. The paper is generally well written and easy to follow. The authors used extensive bioinformatics to describe their data and that is great and well presented. Nevertheless, except from insulin signalling, one may regret that detailed analysis of the different metabolic pathways or signalling cascades that are concerned with differentially expressed genes is presented only as supplementary data, although they are discussed in the "Discussion" section. I would suggest figures 7 and 9 to be presented in the main results (with corresponding description in the Results section. Then, Figures 1 and 3 could be kept as supplementary data. In addition, although adequate restrictions due to the fact that mRNAs do not necessarily reflect protein and pathway activities should be evoked, the authors have here a gold mine, with data in three tissues at the same time in the same bears, to propose possible "metabolic crosstalks" between organs during hibernation.

I raise hereafter minor concerns.

Minor concerns:

1. General:

Same remarks as in general comments about a more detailed analysis of the data.

Because the authors are focusing on insulin signaling, it would have been great to deepen the analysis with complementary techniques (western-blot, enzymatic activities.....)

2. Abstract:

The abstract remains vague regarding the results and I personally prefer when clearly reported results

support clearly stated conclusions or hypotheses.

3. Introduction:

Line 54: please add a reference

Line 58-60: the authors should precise what is known instead of just mentioning that our understanding is incomplete to date

4. Results:

Line 74: delete "were" ?

Lines 77-80: the comparison only considers N=2 bears here and this should be mentioned as a possible limitation.

Figure 3: as far as I understood, this figure presents adipose data. This should be specified in the figure legend.

Supplementary Materials: it would be nice to have the names and titles of supplementary tables clearly mentioned in corresponding files.

General comment: with such unique datasets, it would have been super great to have more details about the different metabolic pathways or signalling cascades that are concerned with differentially expressed genes in the Results section and not only as supplementary data.

5. Discussion:

Lines 280-282: The authors should temper their conclusion, as gene expression does not necessarily reflect protein and pathway activities.

6. Methods:

Line 448: for transparency, this is highly appreciated that the authors give their code. All authors should behave this way. However, I was unable to find the code dedicated to this paper by following the given link. Maybe not implemented yet?

Reviewer #3 (Remarks to the Author):

This is an interesting study looking at the transcriptomes of bears in the spring, fall and winter, representing arousal from hibernation, entrance into hibernation, and torpor. The three organs chosen (adipose, liver, and skeletal muscle) are important metabolically and with respect to muscle atrophy during immobility.

Introduction

The authors do not cite any previous work done analyzing the adipose and skeletal muscle transcriptomes in hibernating ground squirrels. These papers (along with those on other species mentioned in this paper) should be cited in the introduction to illustrate that previous work has been done in this field. The results of the author's study should also be compared to the ground squirrel papers in the discussion as is done for the lemur and seal pups.

Molecular interactions underpinning the phenotype of hibernation in mammals. Andrews MT. *J Exp Biol.* 2019 Jan 25;222(Pt 2). pii: jeb160606. doi: 10.1242/jeb.160606. Review. PMID: 30683731

Gene expression changes controlling distinct adaptations in the heart and skeletal muscle of a hibernating mammal. Vermillion KL, Anderson KJ, Hampton M, Andrews MT. *Physiol Genomics.* 2015 Mar;47(3):58-74. doi: 10.1152/physiolgenomics.00108.2014. Epub 2015 Jan 8. PMID: 25572546

Deep sequencing the transcriptome reveals seasonal adaptive mechanisms in a hibernating mammal. Hampton M, Melvin RG, Kendall AH, Kirkpatrick BR, Peterson N, Andrews MT. *PLoS One.*

2011;6(10):e27021. doi: 10.1371/journal.pone.0027021. Epub 2011 Oct 28. PMID: 22046435

Line 43: Given how the discussion goes into detail on insulin resistance associated with obesity in liver, muscle and adipose, this article would be good to work into the introduction or discussion. It clearly explains how these three organs are affected in obese humans and would strengthen the claims made by the authors.

Korenblat KM, Fabbrini E, Mohammed BS, Klein S. Liver, muscle, and adipose tissue insulin action is directly related to intrahepatic triglyceride content in obese subjects. *Gastroenterology*. 2008;134(5):1369–1375. doi:10.1053/j.gastro.2008.01.075

Results

Lines 71-73: The codes used for different samples were not clear until I searched for them and found them in a figure. One solution would be to add the word sample in front of at least the first example, i.e. (sample RAHi).

Line 121: Do the authors have any data to support that the animals are in hyperphagia in September? For example, were food consumption or body mass measured? If it is common knowledge in the bear research community that bears eat more in the fall then this should be cited in the introduction, for example – Compared to May animals, September bears consume 50% more calories and gain 25% more body mass (I made these numbers up). This would help the reader better understand the magnitude of hyperphagia and gain in body weight. This hyperphagia is in stark contrast to the behavior seen in ground squirrels as described below.

Line 151: Also summarize 13-lined ground squirrel transcriptome results here.

Discussion

Bears appear to be different from ground squirrels which become hypophagic just before entering hibernation (see figure 1 in the reference below). It would be good to compare and contrast these hibernation strategies. Again, data showing food consumption patterns in bears, or a reference to an article that does this would strengthen the discussion. The picture in Figure 1 is a good qualitative example, but not of much use scientifically. One possible explanation if the feeding behaviors are different would be that squirrels' body temperature in torpor drop to 4-8C while bears show much shallower torpor. So food in the gut of a squirrel may not be metabolized while that in a bear could, so emptying the GI tract in a squirrel is more important than in a bear. Food left in the GI tract of a squirrel could rot causing an infection.

Hypothalamic gene expression underlying pre-hibernation satiety. Schwartz C, Hampton M, Andrews MT. *Genes Brain Behav*. 2015 Mar;14(3):310-8. doi: 10.1111/gbb.12199. Epub 2015 Feb 17. PMID: 25640202

Line 148: As mentioned in the introduction the results of this study should be compared to those of ground squirrels for the adipose and muscle transcriptomes. Some of the results are very similar with regards to genes used in fat metabolism.

With the dramatic increase in body mass in bears leading up to hibernation, do they develop fatty liver as is seen in obese humans? This could also indicate some resistance to this process in bears.

Methods

Line 300: Please indicate how these bears were housed, ability to move and exercise, (I assume they were not in the wild) and their access to food in May and September (ad libitum?). If food intake was measured please describe the process.

Line 395: The comparison across years may not be valid given the small sample size of two animals being compared.

Figure 1 is eye catching, but not of much value scientifically. A graphical representation showing body temperature, body weight, and food consumption over the course of a year would be much more informative.

Reviewers' comments:

Reviewer #1 (Remarks to the Author):

The manuscript by Jansen et al. uses RNAseq to examine changes in gene expression patterns between summer active bears, summer active hyperphagic bears, and hibernating bears. The strength of the manuscript is its use of non-biased methods to provide a more complete picture of the molecular events that underlie the hibernation phenotype (it's also important that they have measured at least a couple of summer states [sampling on post-hibernation bears that have increased metabolism but not yet fed would have been illuminating and it's unfortunate this group was not included]).

We agree that the post-hibernation pre-feeding bears would be quite interesting to look at. This time point is especially important for determining the changes in gene expression during the hibernation reversal state. We do not currently know when the expression changes shift from hibernation-like to emerging from hibernation and there is likely a gradient, with an unknown duration, which may vary among bears. We are considering when to sample in the future.

Additionally, because the sample design involves repeated measures, individual variability is accounted for. The manuscript links changes in gene expression to medical disorders (with a focus on insulin-resistance), which should provide broad appeal. The most interesting question in terms of insulin resistance in bears is how the process is reversible, but unfortunately this is beyond the scope of the current study. Many of this study's findings align with prior studies of gene expression during hibernation in bears (citations 5 & 6 in the manuscript), though the design of the current study is more elegant and the number of genes investigated is much larger due to the newer approaches employed. Overall, this manuscript has a similar issue as many other the hibernation genomics papers – the results of gene expression studies are contributing very detailed knowledge regarding processes that have long been known (e.g., that hibernators exhibit insulin resistance – reviewed by Martin 2008 Diabetes and Vascular Disease Research, 5, 76-8) but the question is how does this genomic data move us closer to using this animal model to solve this underlying medical issue) – this is the key element that seems to be lacking here – some form of a path forward, rather than simply a much more detailed understanding of the molecular mechanisms involved in the model organism.

We are encouraged that the reviewer found our study “more elegant” than previous microarray studies and noted that through this design we have achieved a much more detailed understanding of molecular mechanisms involved in hibernation in grizzly bears. This understanding is important for comparing hibernators, such as to ground squirrels identified by reviewer 3. We have added to our discussion the need for future proteomic and enzymatic functional studies. Additionally, we have highlighted the need to sample additional timepoints throughout the year, especially in light of the seasonal

gradient we observed in adipose. Finally, we have included new analyses that highlight the genes whose expression is correlated between tissues, which opens future research into serum factor changes that may regulate multiple metabolically active tissues during hibernation. Given that lipid, glucose and ketone metabolism represent the most obvious differences between hibernating bears and fasting humans or humans with Type 2 diabetes it seems that future research should focus on the details of that regulation

Specific comments:

Line 64: Instead of “small subset” – just state this was done in two animals.

We have revised to “two bears” and clarified that only hibernation was sampled in the two consecutive years.

Line 84: “active plus hyperphagia” is somewhat confusing

We have reworded to “active and hyperphagia”

Lines 148-152: This comparison is a little too simple – you are comparing 3 times as many genes in the bear study compared to the Lemur study and sample size is 6 per group in bears vs. 4 per group in lemurs – thus, some of the differences in # of “significantly” DE genes may simply reflect differences in power between experiments. I would think there are more quantitative ways to make this comparison.

We subsampled our bears to match the sample size of the wild dwarf lemur study to make the comparison more accurate. We also provide absolute numbers from that and the lemur study. The revised sentence is: “However, even when down-sampling our data to match sample sizes within these studies there were dramatic differences in the number of DE genes between the species (lemur 337 DE genes of 10,745 genes total vs. bears with 3,446 DE genes of 17,564 total).”

Line 174-176: “Some genes in the insulin metabolic pathway also have multiple paralogs. Of these gene families, including PIK3, PP2A, and PKC, we found variability in differential expression of paralogs in adipose.” This is unclear – restate this in another way

We have removed this sentence.

Line 186: Don’t use the term “starvation” – replace with “fasting” throughout the manuscript

We have removed the term “starvation” from the manuscript.

Lines 189: “Together, the results in hibernating bears add support to the notion that insulin resistance in adipose plays an important role in energy homeostasis and metabolic disease.” I find this sentence confusing – how does the hibernating bear results inform about the role of adipose in metabolic disease? The hope might be to provide insight into how insulin sensitivity is reversible, but it’s hard to say what underlies this reversible phenotype based off of gene expression data.

We have revised this statement to: “Together, the results of reversible insulin resistance in bears may be helpful in shedding light on the underlying mechanisms involved and thereby aid in the development of treatments for human metabolic disease.”

Lines 287-290: “Based on the extensive changes in tissue expression profiles that occur between active, hyperphagia and hibernation states in grizzly bears, our results suggest a far more complex regulation of cellular function than could be induced by a fast acting “hibernation induction trigger” sought by previous investigators” citation 41 - This seems a bizarre (and old – 1987) citation to use that fails to describe the current view of hibernation research. I agree entirely with the sentiment – but I think this sentiment is shared among the majority of hibernation researchers and seems off-target with the scope of most of the manuscript.

We have removed the part of the sentence about the hibernation trigger.

Supplemental Fig 1 – This isn’t really a test of how representative the tissue is for that state – at least some comparison’s between other states would be helpful (i.e., I expect expression may be highly correlated within tissues, even across states). I don’t doubt that it is representative, but I think there are better ways to illustrate this.

We agree with the reviewer that including comparisons between other states would be helpful. Therefore, we have replaced Supplemental Figure 1 with comparisons between all states within tissues for the two bears sampled during hibernation in the second year.

Supplemental Fig 2 is more informative in this regard (could simply show the different years with different shapes).

We have added the samples from the additional hibernation timepoint to Supplemental Figure 2.

Supplemental Fig 3 – there is no explanation of the different shapes (square vs circle) in the leftmost panels.

We apologize for missing that in the legend. We have included the following in the figure legend: “For the MDS plots, the samples are colored by season: active (red), hyperphagia (yellow), hibernation (blue), shapes differentiate females (circle) and male

(square).”

Supplementary Fig 6 – I assume that black dots were found to be differentially expressed by Fedorov et al. but not in the current study? Are there also genes found to be DE in the present study that were not DE in Fedorov et al?

The black dots are genes that were differentially expressed in Fedorov but not in our study. We have included the following sentence in the figure legend: “Genes that were significantly differentially expressed in Fedorov et al. (2011)¹ but not in our study are in black.” Unfortunately, Fedorov only reports significantly differentially expressed genes so we were only able to compare to that set of genes.

Reviewer #2 (Remarks to the Author):

Communications Biology: Manuscript COMMSBIO-19-0479

Hibernation induces widespread transcriptional remodeling in metabolic tissues of the grizzly bear
Jansen et al.

SYNOPSIS

This study performed RNA-sequencing of adipose, liver, and skeletal muscle of the brown bear. Gene expression was compared among the active, hyperphagia, and hibernation states, extensive changes being shown to occur during hibernation. By using extensive bioinformatics, the authors describe global differences between tissues and/or seasons. Then they focus on insulin signaling, showing that their data would be in line with insulin resistance during hibernation. Changes related to the intermediary metabolism and protein balance are also briefly discussed and appear to be in line with muscle preservation in hibernating animals. The authors conclude that their results may influence positively human health in relation with metabolism, insulin sensitivity, and muscle atrophy.

General comments:

The work appears to have been done carefully and the article presents unique datasets that extend far beyond existing ones. Congrats to the authors. The paper is generally well written and easy to follow. The authors used extensive bioinformatics to describe their data and that is great and well presented. Nevertheless, except from insulin signalling, one may regret that detailed analysis of the different metabolic pathways or signalling cascades that are concerned with differentially expressed genes is presented only as supplementary data, although they are discussed in the “Discussion” section. I would suggest figures 7 and 9 to be presented in the main results (with corresponding description in the Results section).

We have moved supplementary figures 7 and 9 to the main text and significantly increased the results and discussion to include much more general findings about metabolism, muscle atrophy, urea, etc.

Then, Figures 1 and 3 could be kept as supplementary data.

We have added the supplementary figures to the main text without sacrificing existing main text figures (Communications Biology allows 10 display items in the main text for original research articles).

In addition, although adequate restrictions due to the fact that mRNAs do not necessarily reflect protein and pathway activities should be evoked, the authors have here a gold mine, with data in three tissues at the same time in the same bears, to propose possible “metabolic crosstalks” between organs during hibernation.

We have included several sentences in the discussion about the limitations of only investigating transcriptional changes.

We have also included additional results focusing on the genes that are differentially expressed in the same direction in all three tissues, possibility indicating shared regulatory mechanisms and possible “metabolic crosstalk”.

I raise hereafter minor concerns.

Minor concerns:

1. General:

Same remarks as in general comments about a more detailed analysis of the data. Because the authors are focusing on insulin signaling, it would have been great to deepen the analysis with complementary techniques (western-blot, enzymatic activities.....)

Our findings are consistent with other *in vitro* and *in vivo* studies in bears conducted by both co-authors and other groups. We are developing a list of genes to pursue further but this was outside the scope of this initial study. Considering the current interest in hibernation biology, we wanted to publish this initial possible “gold mine” for the research community to access.

2. Abstract:

The abstract remains vague regarding the results and I personally prefer when clearly reported results support clearly stated conclusions or hypotheses.

We agree and we have completely rewritten the abstract.

3. Introduction:

Line 54: please add a reference

We have added references to the sentence.

Line 58-60: the authors should precise what is known instead of just mentioning that our understanding is incomplete to date

We have added more details about what is known from earlier studies to both the introduction and discussion.

4. Results:

Line 74: delete “were” ?

Removed

Lines 77-80: the comparison only considers N=2 bears here and this should be mentioned as a possible limitation.

We have added the actual number of bears to the main text.

Figure 3: as far as I understood, this figure presents adipose data. This should be specified in the figure legend.

Thank you, we have fixed the figure legend.

Supplementary Materials: it would be nice to have the names and titles of supplementary tables clearly mentioned in corresponding files.

We added the names and titles of each supplementary table to the first worksheet in each excel spreadsheet.

General comment: with such unique datasets, it would have been super great to have more details about the different metabolic pathways or signalling cascades that are concerned with differentially expressed genes in the Results section and not only as supplementary data.

We have added substantial text to the discussion to highlight additional metabolic pathways and signaling cascades that are differentially expressed between hibernating and non-hibernating states.

5. Discussion:

Lines 280-282: The authors should temper their conclusion, as gene expression does not necessarily reflect protein and pathway activities.

We have added text to the discussion to highlight that gene expression may not reflect protein and pathway activities and these are important next steps in hibernation studies.

6. Methods:

Line 448: for transparency, this is highly appreciated that the authors give their code. All authors should behave this way. However, I was unable to find the code dedicated to this paper by following the given link. Maybe not implemented yet?

We agree and are supportive of open science. The code and raw data (BioProject) will be made available upon publication.

Reviewer #3 (Remarks to the Author):

This is an interesting study looking at the transcriptomes of bears in the spring, fall and winter, representing arousal from hibernation, entrance into hibernation, and torpor. The three organs chosen (adipose, liver, and skeletal muscle) are important metabolically and with respect to muscle atrophy during immobility.

Introduction

The authors do not cite any previous work done analyzing the adipose and skeletal muscle transcriptomes in hibernating ground squirrels. These papers (along with those on other species mentioned in this paper) should be cited in the introduction to illustrate that previous work has been done in this field. The results of the author's study should also be compared to the ground squirrel papers in the discussion as is done for the lemur and seal pups.

Molecular interactions underpinning the phenotype of hibernation in mammals. Andrews MT. J Exp Biol. 2019 Jan 25;222(Pt 2). pii: jeb160606. doi: 10.1242/jeb.160606.

Review. PMID: 30683731

Gene expression changes controlling distinct adaptations in the heart and skeletal muscle of a hibernating mammal. Vermillion KL, Anderson KJ, Hampton M, Andrews MT. Physiol Genomics. 2015 Mar;47(3):58-74. doi:

10.1152/physiolgenomics.00108.2014. Epub 2015 Jan 8. PMID: 25572546

Deep sequencing the transcriptome reveals seasonal adaptive mechanisms in a hibernating mammal. Hampton M, Melvin RG, Kendall AH, Kirkpatrick BR, Peterson N, Andrews MT. PLoS One. 2011;6(10):e27021. doi: 10.1371/journal.pone.0027021. Epub 2011 Oct 28. PMID: 22046435

Thank you for these helpful suggestions. Our intent was not to diminish the importance of important squirrel work but rather to focus on direct comparisons to bears.

Nevertheless, we completely agree that this approach limited the scope of our findings and failed to provide an important comparative context. Therefore, we have now extensively rewritten the introduction and discussion to include more of the rodent data.

Line 43: Given how the discussion goes into detail on insulin resistance associated with obesity in liver, muscle and adipose, this article would be good to work into the introduction or discussion. It clearly explains how these three organs are affected in obese humans and would strengthen the claims made by the authors.

Korenblat KM, Fabbrini E, Mohammed BS, Klein S. Liver, muscle, and adipose tissue insulin action is directly related to intrahepatic triglyceride content in obese subjects. *Gastroenterology*. 2008;134(5):1369–1375. doi:10.1053/j.gastro.2008.01.075

Thank you, we have added the reference to the introduction.

Results

Lines 71-73: The codes used for different samples were not clear until I searched for them and found them in a figure. One solution would be to add the word sample in front of at least the first example, i.e. (sample RAHi).

We removed the sample codes from the main text and they are now included in the supplement with a clear description for each code.

Line 121: Do the authors have any data to support that the animals are in hyperphagia in September? For example, were food consumption or body mass measured? If it is common knowledge in the bear research community that bears eat more in the fall then this should be cited in the introduction, for example – Compared to May animals, September bears consume 50% more calories and gain 25% more body mass (I made these numbers up). This would help the reader better understand the magnitude of hyperphagia and gain in body weight. This hyperphagia is in stark contrast to the behavior seen in ground squirrels as described below.

Body mass and body temperature were measured during the three sampling periods. These results have added to panel B in Figure 1. We have added several appropriate references to the statement about hyperphagia and the feeding regimen used (see Rigano et al., 2017).

Line 151: Also summarize 13-lined ground squirrel transcriptome results here.

We have summarized 13-lined ground squirrel transcriptome results in the Discussion.

Discussion

Bears appear to be different from ground squirrels which become hypophagic just

before entering hibernation (see figure 1 in the reference below). It would be good to compare and contrast these hibernation strategies. Again, data showing food consumption patterns in bears, or a reference to an article that does this would strengthen the discussion.

We have added additional references and text about differences between small hibernators and brown bears throughout the manuscript.

The picture in Figure 1 is a good qualitative example, but not of much use scientifically. One possible explanation if the feeding behaviors are different would be that squirrels' body temperature in torpor drop to 4-8C while bears show much shallower torpor. So food in the gut of a squirrel may not be metabolized while that in a bear could, so emptying the GI tract in a squirrel is more important than in a bear. Food left in the GI tract of a squirrel could rot causing an infection.

Hypothalamic gene expression underlying pre-hibernation satiety. Schwartz C, Hampton M, Andrews MT. Genes Brain Behav. 2015 Mar;14(3):310-8. doi: 10.1111/gbb.12199. Epub 2015 Feb 17. PMID: 25640202

The bear gut is empty prior to going into hibernation (personal observation by co-author Charles Robbins).

Line 148: As mentioned in the introduction the results of this study should be compared to those of ground squirrels for the adipose and muscle transcriptomes. Some of the results are very similar with regards to genes used in fat metabolism.

See comment above, we have added these comparisons.

With the dramatic increase in body mass in bears leading up to hibernation, do they develop fatty liver as is seen in obese humans? This could also indicate some resistance to this process in bears.

We are not aware of any studies demonstrating fatty liver in hibernating bears.

Of note, co-author Charles Robbins and a graduate student were able to produce fatty livers in the bears several years ago by feeding very high fat, low protein diets. It coincided with a time that liver biopsies were occurring, and the liver samples looked white as if they were actually adipose samples. A pathologist made slides and commented on the fact that they would be considered end-stage fatty liver disease in humans. The bears, as in so many of these metabolic states, cleared the fat out of their livers during hibernation or when put back on more normal diets without any remaining cellular damage. So, fatty livers in bears can be created experimentally, but does not happen on normal diets.

Methods

Line 300: Please indicate how these bears were housed, ability to move and exercise, (I assume they were not in the wild) and their access to food in May and September (ad libitum?). If food intake was measured please describe the process.

Specific details regarding housing, diet, and care of bears can be found in Rigano et al. (2017). We have added this statement to the methods.

Line 395: The comparison across years may not be valid given the small sample size of two animals being compared.

We are limited by the number of bears that were resampled and while the sample size is small, the number of genes that are being compared is large. Moreover, the results are largely descriptive due to the limited sample size.

Figure 1 is eye catching, but not of much value scientifically. A graphical representation showing body temperature, body weight, and food consumption over the course of a year would be much more informative.

We have added a panel to Figure 1 with body temperature and body mass from the three sampling points.

REVIEWERS' COMMENTS:

Reviewer #1 (Remarks to the Author):

The authors have done a good job in revising their manuscript - thanks for the very clear responses to reviewer comments. I have no further concerns.

Reviewer #2 (Remarks to the Author):

This study performed RNA-sequencing of adipose, liver, and skeletal muscle of the brown bear at different periods of its annual cycle. With the changes made according to reviewer suggestions, the results now clearly support metabolic adjustments during hibernation. This, in turn, provide information that may be of importance in relation to various disease states, including for example insulin resistance and muscle atrophy.

I would like to thank the authors, who made a thorough revision of their manuscript. The messages are now clearly supported by a more detailed analysis of DE genes. I do not have remaining concerns.

Reviewer #3 (Remarks to the Author):

This paper is a valuable addition to the body of work on transcriptomes from different tissues, time points, and species of hibernating mammals. The authors have addressed all of my comments and concerns in their revision. The statistical analysis are solid. The additional comparisons to other species put their work in a broader context that will be of interest to researchers in a variety of disciplines.